# Fundamental Movement Skills and Sports Skills: Testing a Path Model

**DOI:** 10.3390/sports13070211

**Published:** 2025-06-27

**Authors:** Fernando Garbeloto, Sara Pereira, Eduardo Guimarães, José Maia, Go Tani

**Affiliations:** 1Motor Behavior Laboratory, School of Physical Education and Sports, University of São Paulo, São Paulo 05508-030, Brazil; gotani@usp.br; 2Centre of Research, Education, Innovation and Intervention in Sport (CIFI2D), Faculty of Sport, University of Porto, 4200-450 Porto, Portugal; sarasp@fade.up.pt (S.P.); eguimaraes@fade.up.pt (E.G.); jmaia@fade.up.pt (J.M.)

**Keywords:** longitudinal study, intervention program, children

## Abstract

This study examined the temporal relationship between fundamental movement skills (FMSs) and sport-specific skills (SSSs) in children aged 7 to 10. Based on the premise that FMSs are the basis for sport skills, we implemented a 10-week intervention program targeting two FMSs (running and stationary dribbling) and one SSS (speed dribbling), followed by immediate and long-term assessments. Using a path-modeling approach, we tested two models: one examining whether FMSs were associated with sport skill performance at the same time point and another exploring whether this influence emerged over time. Results revealed significant FMS and SSS improvements immediately after the intervention program. However, significant associations between the FMSs and SSS emerged only at later time points (8 to 20 months post-intervention), suggesting the delayed influence of the FMSs on the SSS. These findings support that while FMSs are essential for developing more complex skills, their effect may not be immediately observable, emphasizing the importance of long-term follow-up. The results also align with theoretical models contending that proficiency in FMS and sustained practice opportunities are key to integrating fundamental and sport-specific motor skills and may represent an important foundation for public health initiatives advocating early FMS interventions as a strategy to promote lifelong physical activity and sustained engagement in sports.

## 1. Introduction

Child development is a complex process akin to an evolving diamond comprising physical, cognitive, affective, social, and motor facets [1]. Within the motor facet, the study of fundamental movement skills (FMSs) is far-reaching [2,3,4], mainly due to the evidence linking FMS proficiency to increased physical activity levels [5,6] and to positive associations with health, well-being, and BMI [7,8].

In addition to being associated with various health and well-being factors throughout childhood and adolescence—including its potential to counteract one of the major issues of this century, physical inactivity [6,7,8,9]—descriptive heuristic models suggest that FMSs are considered the foundation for the development of more complex skills, such as sport-specific skills (SSSs) [10,11].

Despite the relevance of descriptive models in the field of motor development, most are grounded in sequential perspectives and lack explicit reference to the mechanisms underlying change. One proposal that helps conceptualize these mechanisms is offered by Tani [12], who describes motor development as a hierarchically organized process aimed at forming increasingly complex motor patterns. Within this framework, the acquisition of FMS functions as basic motor patterns that, over time, become components for the acquisition of more complex motor actions, such as SSSs.

For this progression to occur, the child must be able to adjust key motor parameters of the motor patterns—such as force, speed, and direction—without compromising their efficiency. This flexibility enables the functional dismantling of the original FMS, allowing them to be combined into more sophisticated motor patterns.

For example, once a child masters the FMS of running and bouncing a ball, these motor patterns can be used as components of more complex motor patterns—such as speed dribbling. Importantly, this hierarchically organized process is not a simple additive process of components. It involves a reorganization process in which the flexibility of the components plays a crucial role. As the motor development unfolds, what was once a whole (e.g., speed dribbling) may later become a component of an even more complex motor pattern, such as executing a basketball layup. In this hierarchical process, the whole becomes a part, and parts are reorganized to form new wholes [13].

Crucially, this hierarchical process is far from trivial. Using FMSs as building blocks for more elaborate skills requires flexibility. The learner must be able to modify performance parameters (e.g., force, speed, and direction) and adapt each component of the FMS to accommodate new task demands. In other words, the motor pattern of the original FMS must be partially “deconstructed” or altered to allow for functional reorganization into new motor patterns. Thus, the progression from FMSs to SSSs is better understood as a process of increasing behavioral diversity (expanding the range of motor elements by parameters modification) and complexity (integrating these elements into cohesive new patterns) [14].

In this context, both the passage of time and the level of proficiency in FMSs appear to be critical factors when examining their influence on the development of SSSs. However, as highlighted in a recent narrative review [15], the relationship between FMSs and SSSs remains largely underexplored. The review identified only five studies that have directly investigated the connection between FMS and SSS, indicating a significant gap in the literature—particularly regarding how this relationship unfolds over time.

The key findings from those studies include the following: (i) a consistent relationship between FMSs and SSSs that share similar movement patterns, such as the countermovement jump and the high jump [16,17,18,19,20]; (ii) evidence that children with low FMS proficiency struggle to acquire SSSs due to the presence of a proficiency barrier [16,17,19,20]; and (iii) indications that the mastery of specific FMS components (e.g., arm movement during running) is a prerequisite for the development of certain SSSs [17,19,20]. For instance, one study demonstrated that mastery of arm movement while running was essential for performing high-speed basketball dribbling [19].

Despite the acknowledged importance of previous studies, a key gap remains regarding the temporal dynamics of the relationship between FMS and the development of SSS. This raises a critical question: Do improvements in FMS lead to immediate gains in sport skills performance, or do they function as a foundational layer that supports future gains, indicating a time-lagged effect? To investigate this, it is necessary to move beyond cross-sectional associations. That is, intervention studies that promote improvements in FMS should be carried out. Then, the assessment of whether these changes translate into the subsequent development of SSS should be performed accordingly. Moreover, considering motor development as a process of relatively permanent changes in motor behavior [21], researchers must examine the sustainability of intervention effects over time and how this persistence might influence the relationship between FMS and SSS.

A short-term increase in performance does not necessarily imply a persistent developmental change, i.e., a true change. Therefore, assessing performance at multiple time points after the intervention, i.e., a follow-up, enables researchers to evaluate the stability of its effects, as well as to identify any delayed influence that FMS improvements may have on SSS performance. This distinction is decisive: if the relationship between FMS and SSS strengthens or changes over time, it suggests that time is a mediating factor, not only in the acquisition of new skills, but also in the improvement of performance. For this reason, and in our view, at least two post-intervention assessments—one immediately after the program and another at a follow-up point—are essential to capture both short-term outcomes and longer-term developmental trajectories of motor skill integration.

To address this issue, we designed a study based on the following key methodological components: (i) a baseline assessment, (ii) a targeted intervention aimed at improving FMS, and (iii) both immediate and delayed post-intervention assessments of FMS and SSS performance. Such an approach allows us to examine not only whether gains in two FMS (running and stationary dribbling) correlate with improvements in one SSS (speed dribbling) but also if, when, and how these associations emerge. Thus, we formulated the following research questions: (i) Do different levels of FMS proficiency influence the development of SSS over time? (ii) How do specific features of the intervention program—such as its duration, content focus, and integration into regular PE lessons—influence the dynamics between FMS and SSS development? (iii) Does the impact of FMS proficiency on SSS performance emerge immediately, or does it follow a delayed developmental trajectory?

## 2. Materials and Methods

### 2.1. Sample

The data for this study come from a more encompassing research project aiming to investigate the effects of two different intervention programs on the development of FMSs and SSSs across time. For the present paper, we used data from 59 children (29 girls, 30 boys) aged 7–10 years (mean age 8.39 ± 1.09 yrs) at baseline. All children were enrolled in a private school in the city of São Paulo, Brazil.

In the first wave of data collection, children were enrolled in grades 2 to 5 (ages 6 to 10), whereas in the last wave (fourth data collection), they were enrolled in grades 4 to 7 (ages 8 to 12). All children participated in mandatory Physical Education (PE) classes twice a week (class duration = 40 min). They were invited to join the study after receiving a consent form, which their parents or legal guardians signed. Those who had any physical or intellectual disability referenced by their PE teacher that could interfere with or hinder motor assessments were excluded. The project was approved by the University of São Paulo Institutional Review Board (CAAE: 66020517.0.0000.5391).

### 2.2. Intervention Program

An intervention program lasting approximately two months was conducted to investigate whether improvements in FMS proficiency influence SSS development, as well as a follow-up. Figure 1 provides project design template.

The intervention occurred between March and April 2016, consisting of 10 consecutive lessons held once a week, each lasting 40 min, during one of the two weekly PE classes scheduled in the school curriculum. During the other class of the week, the PE teacher continued with the school’s planned content (without any involvement from the research team or using any of the content related to the pedagogical approach designed initially for the study). Only one PE teacher, responsible for children in grades 2 to 5, was also in charge of implementing the intervention program. Before the start of the program, the research project leader met with the PE teacher to discuss how to implement the intervention (e.g., how to give movement instructions and the structure of each class) and to help prepare the content for the 10 classes. To monitor pedagogical consistency, the lead researcher attended four of the ten sessions—specifically, the first, fourth, seventh, and tenth classes—to verify adherence to the planned instructional procedures. Additionally, the PE teacher was instructed to contact the lead researcher in case any session deviated from the original plan or if any uncertainties arose during implementation.

Since the intervention occurred during regular PE classes, all children participated, including those who did not return the signed consent form, although their performance was not assessed. Furthermore, regardless of age, the same intervention was applied to all children. Only the materials used (e.g., type and size of the ball) and the goals of the tasks (e.g., the height of the basket) were adjusted according to the age group’s needs.

Each class began with a brief warm-up (~3 min), followed by games and activities designed to encourage children to practice two FMS (running without an object and stationary dribbling) and one sports skill (speed dribbling—running while dribbling a basketball) in different situations (e.g., with or without an opponent, with changes in direction, at varying speeds). According to Tani [12], practicing the same skill in different contexts promotes diversity, which is a required condition for learning new motor skills.

### 2.3. Performance Measures

The Test of Gross Motor Development—Second Edition [22]—was used to assess six FMS: running, hopping, leaping, kicking, catching, and stationary dribbling. Each skill comprises three to five qualitative performance criteria, scored as 1 when the criterion is met and 0 when it is not. All assessments were conducted according to the procedures outlined in the TGMD-2 manual, with performance scores calculated as the sum of the scores from two trials. Although all six FMS were evaluated, for this study, we focused exclusively on the combined raw scores for running and stationary dribbling. The total score ranged from 0 to 16 points, with up to 8 points allocated for proficient performance in each skill, defined as meeting all criteria in both trials of the respective tasks.

The sport skill was evaluated using a valid and reliable checklist developed to assess proficiency levels in speed dribbling [23]. In this procedure, children were required to cover, twice, a distance of 18 m as fast as possible while running and dribbling a basketball with their preferred hand. The checklist consists of nine binary criteria for evaluating the movement, similar to the TGMD-2 scoring system, where each criterion is scored as 1 if observed or 0 if not. This SSS was selected because it integrates running and stationary dribbling (two FMS), and its assessment protocol is aligned with TGMD-2 guidelines, i.e., focus on movement component criteria. The outcome measure of this sports skill is the sum of the components proficiently performed in the two attempts. The values can range from 0 (does not perform any component proficiently) to 18 (performs all components proficiently).

FMS and SSS performance were recorded with a Sony HDR-PJ540 camera (60 Hz), and the assessment procedure was in line with the guidelines described by Santos [23]. All 59 children’s videos were rated by the principal leader of the research project (the first author). To assess intra-rater reliability, 20 videos were re-evaluated one week later in a random order, and agreement was calculated with Cohen’s κ. For TGMD-2 FMS assessments, the Cohen’s κ ranged from 0.75 for leaping to 0.95 for stationary dribbling. Further, the intra-rater agreement was calculated for each of the nine components for speed dribbling, with the Cohen’s κ ranging from 0.73 for component 9 [the ball forward and laterally (not in front of the body)] to 0.91 for component 6 (dominates the ball from the waist down).

### 2.4. Data Collection

One week after the conclusion of the intervention program, the second wave of data collection was performed (between May and June 2016). After completing this data collection, the PE teacher responsible for the classes was instructed to follow his original PE annual plan. During this period, without any intervention from the research members, classes were routinely held twice a week, lasting 40 min each.

Then, the follow-up consisted of two new data collection waves. The first was executed in February 2017, with children enrolled in grades 3 to 6. After the data collection, all children were exposed to all PE classes designed by their PE teachers (without any intervention from the research team) for the remainder of the school year. Finally, to inspect a putative long-term effect of the intervention program on FMS performance (running and stationary dribbling) and SSS performance (basketball speed dribbling), a second data collection procedure was carried out in February 2018 with children in grades 4 to 7 of elementary school. In sum, the research project has four data waves of data collection—before (first wave) and after the intervention (second wave), and two follow-up measures (third and fourth waves).

### 2.5. Statistical Analysis

Basic descriptive statistics include medians and interquartile range due to the skewness of the data distribution, and a Friedman test was used to see if changes were statistically significant. This analysis was performed in SPSS v.27. To investigate whether changes promoted by the intervention program affected the dynamics of the relationship between running, stationary dribbling, and speed dribbling—and to examine whether the influence of FMS proficiency on sport skill performance emerges immediately (Figure 2) or follows a delayed developmental trajectory (Figure 3)—we used a path model within the tradition of structural equation modeling [24,25,26].

The acronym RD represents the sum of the proficient components performed in the FMSs of running and stationary dribbling. Also, Db represents the sum of the proficient components performed in the sports skill (speed dribbling). As depicted in Figure 2, this model comprises two parts: the first is the direct influence of adjacent measures RD_1_ on RD_2_, RD_2_ on RD_3_, and RD_3_ on RD_4_ represented by the auto-regression coefficients β_21RD_,…, β_43RD_, and the same direct effects are expressed in the time influence of adjacent measures of Db_1_ on Db_2_, Db_2_ on Db_3_, and Db_3_ on Db_4_, with their corresponding auto-regression coefficients (β_21Db_, …, β_43Db_). This part of the model reflects the normative stability in the intra-skill developmental process. The second part of the model comprises the coincident influence of RD_2_ on Db_2_, RD_3_ on Db_3_, and RD_4_ on Db_4_ with their corresponding regression coefficients β_Db2,RD2_, β_Db3,RD3_, and β_Db4,RD4_. Finally, e_RD2_ to e_RD4_ and e_Db2_ to e_Db4_ are regression residuals from this system of regression equations, and ζ_2_, ζ_3_, and ζ_4_ are the correlations between residuals. In sum, the hypothesis we are testing in this model (model 1) is that the development trend within each skill (RD and Db) shows stability across children’s proficiency levels and that RD has a direct and coincident effect on Db at each time point. Finally, ρ_RD1,Db1_ is the correlation between the two variables at baseline.

In model 2 (Figure 3), the only difference is that there are no direct effects of RD on Db, but there are lagged effects. Hence, we hypothesize that irrespective of the specific auto-regressive effects describing changes within RD and Db, the main change in this model is that there is a delayed effect of RD developmental levels on Db: β_Db2,RD1_, β_Db3,RD2_, and β_Db4,RD3_.

We relied on EQS 6.2 software [26] to simultaneously estimate the parameters of the path analytical model presented in Figure 2 and Figure 3 using maximum likelihood with a robust solution. The overall goodness-of-fit of each model was assessed with standard statistical indices as advocated [24,25,26,27]: chi-square (χ^2^), Comparative Fit Index (CFI), and Root Mean Square Error of Approximation (RMSEA). In all parameter estimates, an alpha of 5% was used.

## 3. Results

Descriptive statistics across the four time points for running, stationary dribbling, and the speed dribbling are in Table 1, and their graphical display is in Figure 4.

The Friedman test showed significant increases (χ^2^_(3)_ = 26.72, *p* < 0.001) in running and stationary dribbling, as well as in speed dribbling (χ^2^_(3)_ = 22.16, *p* < 0.001).

In order not to overload Figure 5 with all parameter estimates, we just retained those that express the core of the model, i.e., the autoregressive coefficients (in blue), the direct coincident paths (in red), and the correlation between the baseline values of RD_1_ and Bd_1_ (in black). These parameter estimates are standardized. All autoregressive coefficients for RD are statistically significant (*p* < 0.05) but are decreasing in time. For Db, they are also statistically significant (*p* < 0.05), but their magnitude remains relatively similar. Except for the direct path from RD_4_ to Db_4_, the others are not statistically significant (*p* > 0.05). The correlation between RD1 and Db1 is statistically significant (*p* < 0.05). This model shows a marginal fit to the data: χ^2^_SB_ = 28.316, *p* = 0.019; CFI = 0.865; RMSEA = 0.124 (95% CI: 0.048–0.191).

Similarly to the previous model, in model 2 (Figure 6), the measures of global fit are the same, as are the standardized autoregressive parameters (in blue) and the correlation (in black). Only the lagged effect (in red) from RD_3_ to Db_4_ is statistically significant (*p* < 0.05), whereas the other two are not (*p* > 0.05).

## 4. Discussion

The present study addressed the following research questions: (i) Do different levels of two FMSs’ proficiency (running and stationary dribbling) influence the development of one SSS (speed dribbling) over time? (ii) How do specific features of the intervention program—such as its duration, content focus, and integration into regular PE lessons—influence the dynamics between FMS and SSS development? (iii) Does the impact of FMS proficiency on the SSS performance emerge immediately, or does it follow a delayed developmental trajectory? Our overall model fit shows values within the boundary of acceptable fit in these types of models with directly observed variables. This may be linked to a small sample size, a different time lag between data waves, and/or other environmental issues that were not considered in the study. It is also important to note that in these types of models, only variances and covariances are used (stability and coincident/cross-lagged effects) and do not address changes in means across time. We were very careful to not rely on model re-specification, freeing parameters that were not substantively interpretable, even at the cost of increasing model fit statistics. Hence, and within the limits of these considerations, we used a robust estimation procedure to tackle the issue of the sample size and have precise standard errors, which are important for significant testing of all regression effects, correlations, and error covariances. Further, since both models have the same number of degrees of freedom and the same chi-square statistic and other fit indices (CFI and RMSEA), we believe that they can be suitably interpretable within the study framework, and the choice depends on how much different they are in substantive terms and/or in what they suggest in terms of the transitional influence of the FMSs on the SSSs.

Regarding the first question, our findings suggest that different levels of FMS proficiency influence the development of SSSs over time. However, this influence does not appear to be consistent across all time points. As shown in the tested models, the FMS performance (RD) was significantly associated with the SSS performance (Db) only at specific moments. In model 1 (coincident effects), this association was observed exclusively at the final time point (RD_4_→Db_4_). In model 2 (lagged effects), only the path from RD_3_ (time 3) to Db_4_ (time 4) reached statistical significance. These results, showing an association only at specific points in time, may be related to the well-known proficiency barrier. According to Seefeldt [28], children with low FMS proficiency may encounter a proficiency barrier that limits progress towards more complex motor skills. Recent findings support this idea, indicating that children with low levels of FMSs tend to perform poorly in SSSs, even after participating in intervention programs [16,17]. Although the present study did not directly analyze the individual effects of low FMS proficiency, our results revealed a significant improvement in performance between time 1 and 2, suggesting that gains in the FMS (during the intervention program) may influence the development of the SSS over time. However, as we will address later, improvements in FMSs do not appear to translate into immediate gains in the SSS performance. Hence, time seems to be a key factor in consolidating more complex motor skills.

Furthermore, it is important to note that the concept of the proficiency barrier may also apply to the relationship between the FMS proficiency and the physical activity levels. Recent studies have shown that children with very low levels of FMSs are less likely to engage in moderate-to-vigorous physical activity (MVPA) and to meet the World Health Organization (WHO) guidelines for daily physical activity [6,29]. In this context, low FMS proficiency may have broader health implications, as an insufficient MVPA is associated with an increased risk of sedentary behavior and long-term adverse health outcomes.

With regard to the second research question, the results showed that the structure of our intervention program—particularly the deliberate variation in the same FMS (e.g., through changes in force, speed, and direction) and the progressive increase in the task complexity (by combining FMS components)—was effective in enhancing the performance. Despite its relatively short duration (10 sessions), this structured approach led to significant improvements in both FMSs and SSSs, as demonstrated by the Friedman test.

These findings reinforce the existing literature on the effectiveness of structured interventions in promoting FMS development [4,30]. However, the statistical modeling revealed that the relationship between FMSs and SSSs is neither “linear” nor immediate, thus justifying the examination of models with time-lagged effects. However, various motor development heuristic models [1,10] assume a hierarchical and interdependent relationship between FMSs and SSSs, and few explicitly address how and when this influence occurs. Model 1 tested the hypothesis that FMS proficiency would influence SSS performance at concurrent time points. This approach was based on prior studies that reported simultaneous gains in both skill categories following intervention programs [18,31] and on evidence suggesting that children may develop simple and complex skills concurrently from early childhood [32].

Our findings also revealed that the FMS performance showed a lower stability over time, as reflected in decreasing autoregressive coefficients, indicating changes in children’s relative performance rankings. In contrast, the SSS performance had a greater temporal stability. This could be interpreted in two ways: (i) once the SSS is mastered, it tends to remain stable even without further reinforcement, provided by new systematic practice sessions; or (ii) the gains observed between the first and second time points were sufficient to consolidate proficiency levels, thereby maintaining individual performance rank. Further, relative changes in the SSS performance would likely require additional acute stimuli, namely a second intervention program. On the other hand, as FMSs are more frequently practiced in everyday activities (e.g., play, sports, physical education), children may have been exposed to diverse stimuli that helped those with a lower initial FMS performance improve their skills and change their relative position compared to other participants.

Regarding the third research question, our findings do not provide a definitive conclusion as to whether the relationship between FMSs and SSSs emerges concurrently or with a delay, as both models showed equivalent global fit indices. However, it is noteworthy that in both the coincident and lagged models, significant associations between variables were observed only between 8 and 20 months after the intervention, suggesting a possible delayed effect of the FMS improvements on the SSS performance. Results from both models 1 and 2 indicate that the proficiency achieved in RD_2_ (time 2), combined with ongoing practice opportunities (between RD_2_ and RD_3_), may have triggered a process of motor reorganization and diversification that ultimately supported the development of Db, as observed at the final time point. This process aligns with the theoretical perspective of Tani [12], who proposed that, from a hierarchical view of motor development, FMSs function as motor patterns that, over time, become basic components for the acquisition of more complex skills, such as SSS. For this progression to occur, the child must be able to adjust key motor parameters—such as force, speed, and direction—without compromising the efficiency of motor patterns. This flexibility enables a functional dismantling of the original FMSs, allowing for their components to be reorganized into more sophisticated motor patterns.

Based on the hierarchical approach, this process illustrates how elements of running and stationary dribbling patterns were initially consolidated through practice and subsequently integrated to form a more complex skill—speed dribbling in basketball. This dynamic reorganization, in which motor patterns alternate between functioning as autonomous skills and as building blocks of more advanced motor actions, depends on sufficient time and practice conditions. In our intervention program, these conditions—specifically, the promotion of movement variability and the progressive increase in task complexity—were deliberately integrated to support this developmental progression.

In this sense, the initial intervention appears to have worked as a developmental mediator [33], allowing children to achieve a foundational level of motor performance that improved their proficiency and shaped the conditions for integrating previously acquired motor skills into more complex skills. According to Fischer [34], executing a skill at a higher level of complexity (e.g., sport-specific skills) is directly influenced by the proficiency achieved in simpler skills (e.g., FMS). However, for such transitions to occur, two key factors must be present: first, the lower-level skills must reach a sufficient level of performance quality; second, the environment must offer affordances that encourage individuals to explore and connect skills across different levels of complexity. Therefore, the dynamics of motor development depend not only on proficiency but also on the availability of opportunities and the passage of time.

In addition to motor proficiency and practice conditions, perceptual–cognitive components, such as attentional control, and perceptual–motor integration may also moderate the transition from FMSs to SSSs. For example, Mancini et al. [35] demonstrated that training with perception–action devices significantly improved reaction time and quickness in volleyball players, suggesting that perceptual–cognitive abilities can play a key role in refining SSS behaviors. While this perspective was not directly tested in our study, future research should explore how perceptual–cognitive variables interact with motor proficiency in supporting the emergence of complex motor skills.

## 5. Study Limitations and Conclusions

The findings of this study indicate that different levels of proficiency in FMSs influence the development of SSSs over time. Moreover, even with a relatively brief intervention program (10 sessions), significant improvements were observed in two FMSs and one SSS. However, this study has some important limitations. First, the study did not include a control group, and all participants were drawn from a private school in São Paulo, representing a relatively homogeneous cultural and socioeconomic context. These factors make it difficult to determine whether the observed improvements were due solely to the intervention or also influenced by natural developmental processes or greater access to structured physical activity opportunities. This methodological decision was made in order to preserve the ecological validity of the study, as school administrators and PE teachers required that all classes receive the same pedagogical content.

While we recognize the importance of conducting research in real-world school settings, we also acknowledge that the absence of a comparative condition limits the internal validity of our findings, and that the homogeneity of the sample may constrain the external validity and generalizability of the results to more diverse educational contexts. Future studies could benefit from implementing quasi-experimental designs with matched control groups and including participants from a broader range of school settings in order to better account for contextual variation and enhance the robustness of the conclusions.

Another limitation concerns the focus on only two FMSs and one SSS, which may have resulted in some children reaching a performance ceiling, particularly in running, shortly after the intervention. Given that our model depends on individual differences in change over time, a ceiling effect could indeed lead to an underestimation of the influence of running proficiency, both in isolation and in combination with other FMSs (e.g., stationary dribbling) when predicting SSS outcomes. Although the combined FMS score (running + stationary dribbling) helped mitigate this issue to some extent, we recognize that running may have disproportionately contributed to the reduced variance in later waves. Nevertheless, this constraint allowed for detailed longitudinal tracking of skill development. Given the scarcity of longitudinal studies examining FMS development over two years, this research represents a novel and meaningful contribution to understanding the temporal dynamics between FMSs and SSSs in real-world school settings. Notably, the main findings suggest that while immediate improvements were evident, significant associations between the FMSs and the SSS emerged only 8 to 20 months after the intervention, indicating a potential delayed effect of FMSs on SSSs’ development. It is also worth highlighting that the delayed effect of the FMSs on the SSS reinforces the importance of a long-term investment in FMSs development programs within school curricula. Enhancing FMSs may not only support motor proficiency but also contribute to children’s health by promoting adequate physical activity levels, physical fitness, and a healthy lifestyle throughout their lifespan.

## Figures and Tables

**Figure 1 sports-13-00211-f001:**
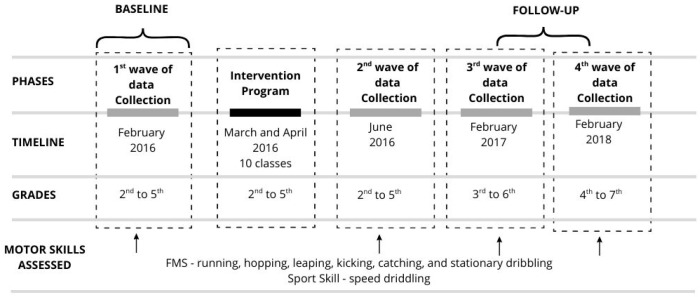
Project design template.

**Figure 2 sports-13-00211-f002:**
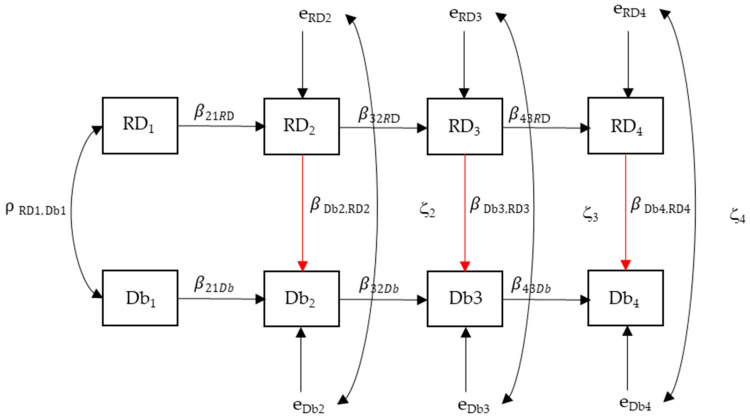
Model 1. The effects of running and stationary dribbling (RD) occur in dribbling at speed (Db) at coincident time points marked in red.

**Figure 3 sports-13-00211-f003:**
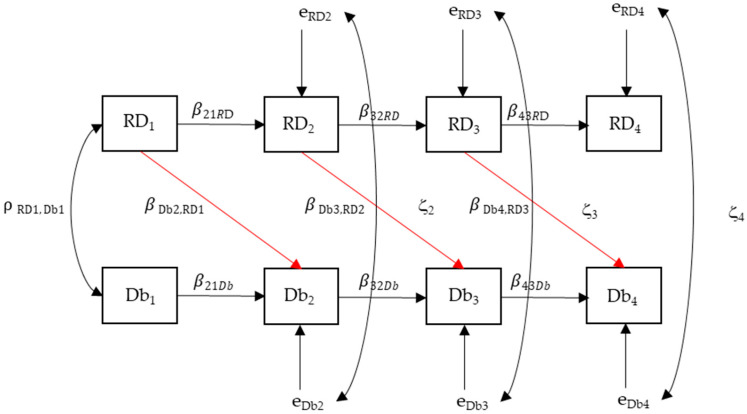
Model 2. First-order autoregressive model for running and stationary dribbling (β_RD_) and speed dribbling (β_Db_), as well as the cross-lagged temporal effects of running and stationary dribbling on speed dribbling (β_Db,RD_) marked in red.

**Figure 4 sports-13-00211-f004:**
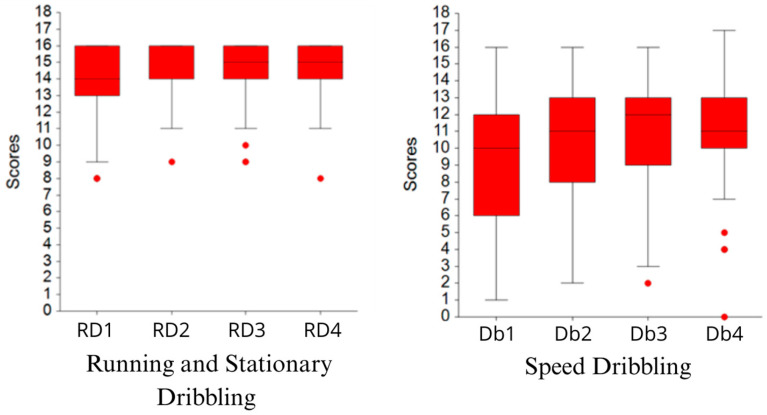
Box-plots for running and stationary dribbling (**left**) as well as for speed dribbling (**right**).

**Figure 5 sports-13-00211-f005:**
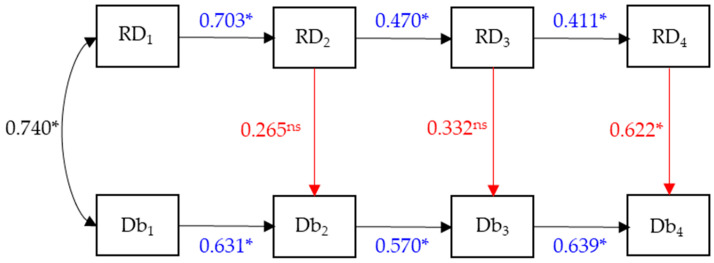
Standardized parameter estimates for model 1. ns = non significant; * = *p* < 0.05. Numbers in blue = Autoregressive parameters. Numbers in red = Direct coincident paths.

**Figure 6 sports-13-00211-f006:**
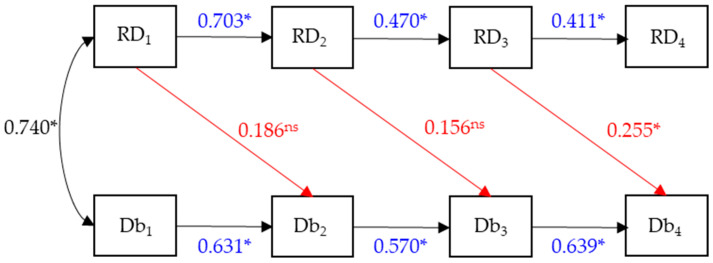
Standardized parameter estimates for model 2. ns = non significant; * = *p* < 0.05. Numbers in blue = Standardized autoregressive parameters. Numbers in red = Lagged effect.

**Table 1 sports-13-00211-t001:** Basic statistics [medians and their 95% confidence intervals (95% CI) and interquartile range] for running and stationary dribbling and speed dribbling across the four data waves.

Variables	Medians	Interquartile Range	95% CI
Running and Stationary Dribbling 1	14	3	13–15
Running and Stationary Dribbling 2	16	2	15–16
Running and Stationary Dribbling 3	15	2	14–16
Running and Stationary Dribbling 4	15	2	14–16
Speed Dribbling 1	10	6	8–11
Speed Dribbling 2	11	5	10–12
Speed Dribbling 3	12	4	10–12
Speed Dribbling 4	11	3	11–12

## Data Availability

The data is a property of the School of Physical Education and Sport, University of S. Paulo, Brazil, and is therefore protected from being freely shared. Yet, all researchers wanting to use the data will have to comply with the School policies. Further, they can address their request to the main author—Fernando Garbeloto.

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
