# Peer review of "Fundamental Movement Skills and Sports Skills: Testing a Path Model"

_sports, 2025, doi:10.3390/sports13070211_

Round 1
Reviewer 1 Report
Comments and Suggestions for Authors
The manuscript titled “Fundamental Movement Skills and Sports Skills: Testing a Path Model” presents an important and well-conceptualized contribution to developmental motor research. The authors aim to evaluate the longitudinal dynamics between fundamental movement skills (FMS) and sport-specific skills (SSS) through a structured intervention and multiple follow-ups using robust path modeling. While the work is methodologically sound and addresses a significant gap in the literature, several aspects should be refined to strengthen its scientific rigor and interpretability.
From a formal perspective, the manuscript is clearly structured, the language is overall fluent, and the argumentation is generally coherent. However, a more concise and varied lexicon would improve the readability. Phrases such as “running and stationary dribbling” or “speed dribbling” are used repetitively throughout, at times making the narrative monotonous. Rephrasing or grouping them under common constructs when appropriate could reduce redundancy. Moreover, it would benefit the reader to clearly distinguish between coincident and lagged effects from the outset, perhaps with a short didactic insert explaining the rationale behind this modeling choice before the results section.
Regarding the scientific content, the intervention design is ecologically valid and well-aligned with practical settings, particularly the integration within regular PE lessons. Nonetheless, the absence of a control group is a non-trivial limitation. While the authors acknowledge this, the justification that all classes had to receive the same pedagogical content, though pragmatic, does not eliminate the need for at least a minimal comparative condition or statistical alternative. Even if ethically or logistically difficult, employing statistical controls for maturation effects or using a matched comparison sample retrospectively might provide a partial solution. In future studies, adopting a quasi-experimental design with matched control classes could resolve this issue without compromising school-wide interventions.
The statistical analyses are well executed. The authors rightly use path modeling to test both coincident and time-lagged relationships, a valuable approach given the developmental nature of the constructs. Nonetheless, the reported fit indices (especially RMSEA values above 0.10) indicate marginal model fit. While this is not uncommon in complex longitudinal models with small samples, it warrants a more thorough discussion. The authors should consider whether alternative specifications (e.g., including error covariances, or testing nonlinear paths) might yield a better model fit. Furthermore, the report could be strengthened by including Akaike or Bayesian Information Criteria (AIC/BIC) to better justify the choice of one model over another.
In the interpretation of the findings, the authors propose that improvements in FMS translate into delayed benefits in SSS. This hypothesis is well supported by the significant path from RD3 to Db4 in Model 2. Nevertheless, the generalization of this conclusion requires caution, given that this was the only significant lagged effect. A more critical reflection on why other paths did not reach significance—even when skill gains were observed—would enhance the credibility of the conclusions. One possibility is that continued environmental stimulation, rather than the initial gains per se, mediated longer-term outcomes.
The discussion section correctly references theories such as Seefeldt’s proficiency barrier and Tani’s developmental mediators, and provides an engaging reflection on motor learning mechanisms. However, it would benefit from further theoretical elaboration concerning the role of perceptual-motor integration and attentional control in the transition from FMS to SSS. I suggest integrating the work by Mancini et al. (2024) — The Impact of Perception–Action Training Devices on Quickness and Reaction Time in Female Volleyball Players, Journal of Functional Morphology and Kinesiology, 9(3), 147. This study supports the idea that perceptual-cognitive components are crucial in refining sport-specific behaviors and may act as moderating variables in the link between foundational and performance skills. Including this perspective would enrich the theoretical discussion and open new directions for future intervention designs.
In terms of limitations, the authors rightly point to the absence of a control group and to the limited scope of the motor skills assessed. It would also be appropriate to comment on the sample's cultural and socioeconomic homogeneity, as all participants were enrolled in a private school in São Paulo. This may limit the external validity of the results, particularly when considering how contextual factors like access to structured sport programs and extracurricular opportunities may affect the generalizability of motor skill development trajectories.
Finally, the bibliography is largely adequate and recent, though somewhat insular. Many of the cited works are authored by the research team or collaborators. While this is understandable in niche fields, including a broader set of international references could strengthen the manuscript’s positioning within the global literature.
In conclusion, this study offers a valuable contribution to the understanding of how fundamental and sport-specific motor skills develop over time. The intervention is well-conceived and the longitudinal approach is particularly commendable. However, revisions in the theoretical framework, statistical interpretation, and stylistic clarity—as well as the integration of new literature such as Mancini et al. (2024)—are recommended to enhance the manuscript's impact and rigor. With these improvements, the article will better serve the field of developmental motor behavior and physical education pedagogy.
Author Response
Dear Reviewer,
We are grateful for the reviewer’s insightful comments, which have contributed to improving the clarity and depth of our work. All corresponding changes have been made in the manuscript and are highlighted in yellow for easy reference.
Reviewer 1
Comments and Suggestions for Authors
The manuscript titled “Fundamental Movement Skills and Sports Skills: Testing a Path Model” presents an important and well-conceptualized contribution to developmental motor research. The authors aim to evaluate the longitudinal dynamics between fundamental movement skills (FMS) and sport-specific skills (SSS) through a structured intervention and multiple follow-ups using robust path modeling. While the work is methodologically sound and addresses a significant gap in the literature, several aspects should be refined to strengthen its scientific rigor and interpretability.
From a formal perspective, the manuscript is clearly structured, the language is overall fluent, and the argumentation is generally coherent. However, a more concise and varied lexicon would improve the readability. Phrases such as “running and stationary dribbling” or “speed dribbling” are used repetitively throughout, at times making the narrative monotonous. Rephrasing or grouping them under common constructs when appropriate could reduce redundancy. Moreover, it would benefit the reader to clearly distinguish between coincident and lagged effects from the outset, perhaps with a short didactic insert explaining the rationale behind this modeling choice before the results section.
Authors answer: We want to thank the reviewer for her/his generous comments and suggestions to our paper that undoubtedly enhanced its quality. Please note also that all occurrences of "fundamental movement skills" and "specific sport skills" have been replaced with their respective abbreviations, "FMS" and "SSS," throughout the text, and that we add a sentence regarding the path models used.
Reviewer: Regarding the scientific content, the intervention design is ecologically valid and well-aligned with practical settings, particularly the integration within regular PE lessons. Nonetheless, the absence of a control group is a non-trivial limitation. While the authors acknowledge this, the justification that all classes had to receive the same pedagogical content, though pragmatic, does not eliminate the need for at least a minimal comparative condition or statistical alternative. Even if ethically or logistically difficult, employing statistical controls for maturation effects or using a matched comparison sample retrospectively might provide a partial solution. In future studies, adopting a quasi-experimental design with matched control classes could resolve this issue without compromising school-wide interventions.
Authors answer: Many thanks for this comment and suggestion. We have expanded our discussion regarding the study limitation and provided additional information that may help readers address similar challenges in future research.
The following was added:
First, the study did not include a control group, and all participants were drawn from a private school in São Paulo, representing a relatively homogeneous cultural and socioeconomic context. These factors make it difficult to determine whether the observed improvements were due solely to the intervention, or were also influenced by the “natural” developmental processes or greater access to structured physical activity opportunities. Yet, our methodological decision was made in order to preserve the ecological validity of the study, as school administrators and PE teachers required that all classes receive the same pedagogical content.
While we recognize the importance of conducting research in real-world school settings, we also acknowledge that the absence of a comparative condition limits the internal validity of our findings, and that the homogeneity of the sample may constrain the external validity and generalizability of the results to more diverse educational contexts. Future studies could benefit from implementing quasi-experimental designs with matched control groups and including participants from a broader range of school settings, in order to better account for contextual variation and enhance the robustness of the conclusions.
Reviewer: The statistical analyses are well executed. The authors rightly use path modeling to test both coincident and time-lagged relationships, a valuable approach given the developmental nature of the constructs. Nonetheless, the reported fit indices (especially RMSEA values above 0.10) indicate marginal model fit. While this is not uncommon in complex longitudinal models with small samples, it warrants a more thorough discussion. The authors should consider whether alternative specifications (e.g., including error covariances, or testing nonlinear paths) might yield a better model fit. Furthermore, the report could be strengthened by including Akaike or Bayesian Information Criteria (AIC/BIC) to better justify the choice of one model over another.
Authors answer: We thank the reviewer for her/his comments on our statistical analysis. In fact, given the small sample size and the time lag between observations, the RMSEA is within the boundary of a reasonable fit, even using a robust estimation procedure. Please note, however, that we followed the general “rules” regarding these types of models and tried to maintain a perspective of model testing, and not follow the ideas of model re-specification that sometimes tend to overparameterize models without a substantive reason. EQS has a series of capabilities regarding model changes. We used them and found that many suggestions were impractical given what the software names “condition code”, i.e., negative variances and/or correlations greater than 1, and others would change the main aim of both models. These are the reasons why we “stick” to our model testing. Please note also that both models had the same degrees of freedom and the same value of Satorra-Bentler Chi-square. This way AIC or BIC would be the same, and so we did not use it. We add a comment on this within the lines of our comment.
The following was added, which also covers the next comment:
Our overall model fit shows values within the boundary of acceptable fit in these types of models with direct observed variables. This may be linked to a small sample size, a different time lag between data waves, and/or other environmental issues that were not considered in the study. It is also important to note that in these types of models, only variances and covariances are used (stability and coincident/cross-lagged effects), and do not address changes in means across time. We were very careful in not relying on model re-specification, freeing parameters that were not substantively interpretable, even at the cost of increasing model fit statistics. Hence, and within the limits of these considerations, we used a robust estimation procedure to tackle the issue of the sample size, and have precise standard-errors which are important for significant testing of all regression effects, correlations, and error covariances. Further, and since both models have the same number of degrees of freedom and the same chi-square statistic and other fit indices (CFI and RMSEA), we believe that they can be suitable interpretable within the study framework, and the choice depends on how much different they are in substantive terms, and/or in what they suggest in terms of transitional influence of FMS on SSS.
Reviewer: In the interpretation of the findings, the authors propose that improvements in FMS translate into delayed benefits in SSS. This hypothesis is well supported by the significant path from RD3 to Db4 in Model 2. Nevertheless, the generalization of this conclusion requires caution, given that this was the only significant lagged effect. A more critical reflection on why other paths did not reach significance—even when skill gains were observed—would enhance the credibility of the conclusions. One possibility is that continued environmental stimulation, rather than the initial gains per se, mediated longer-term outcomes.
Authors answer: Again, we thank the reviewer for this call. Please note, however, that path models deal with variances and covariances, not means. To test for gains, linear or curvilinear, across the four data waves would require a different statistical approach, and to test for the presence of environmental factors as well as putative mediating variables would be something beyond the reach of this paper, and we wonder if there is a published paper that ever tackled such a complex issue.
Reviewer: The discussion section correctly references theories such as Seefeldt’s proficiency barrier and Tani’s developmental mediators, and provides an engaging reflection on motor learning mechanisms. However, it would benefit from further theoretical elaboration concerning the role of perceptual-motor integration and attentional control in the transition from FMS to SSS. I suggest integrating the work by Mancini et al. (2024) — The Impact of Perception–Action Training Devices on Quickness and Reaction Time in Female Volleyball Players, Journal of Functional Morphology and Kinesiology, 9(3), 147. This study supports the idea that perceptual-cognitive components are crucial in refining sport-specific behaviors and may act as moderating variables in the link between foundational and performance skills. Including this perspective would enrich the theoretical discussion and open new directions for future intervention designs.
Authors answer: We truly appreciate your suggestion. The referenced study provides a valuable perspective that contributes to a deeper understanding of the potential moderating role of perceptual-cognitive components in the development of sport-specific skills. We have integrated this reference into our Discussion to enrich the theoretical framework and inform future research directions.
The following was added:
In addition to motor proficiency and practice conditions, perceptual–cognitive components such as attentional control and perceptual–motor integration may also moderate the transition from FMS to SSS. For example, Mancini et al., (2024) demonstrated that training with perception–action devices significantly improved reaction time and quickness in volleyball players, suggesting that perceptual–cognitive abilities can play a key role in refining SSS behaviors. While this perspective was not directly tested in our study, future research should explore how perceptual–cognitive variables interact with motor proficiency in supporting the emergence of complex motor skills.
Reviewer: In terms of limitations, the authors rightly point to the absence of a control group and to the limited scope of the motor skills assessed. It would also be appropriate to comment on the sample's cultural and socioeconomic homogeneity, as all participants were enrolled in a private school in São Paulo. This may limit the external validity of the results, particularly when considering how contextual factors like access to structured sport programs and extracurricular opportunities may affect the generalizability of motor skill development trajectories.
Authors answer: Again, thank you for the suggestion. We agree that the cultural and socioeconomic homogeneity of the sample represents an important limitation. This point has been added to the limitations section to highlight potential constraints on the generalizability of our findings.
First, the study did not include a control group, and all participants were drawn from a private school in São Paulo, representing a relatively homogeneous cultural and socioeconomic context. These factors make it difficult to determine whether the observed improvements were due solely to the intervention or also influenced by natural developmental processes or greater access to structured physical activity opportunities. This methodological decision was made in order to preserve the ecological validity of the study, as school administrators and PE teachers required that all classes receive the same pedagogical content.
While we recognize the importance of conducting research in real-world school settings, we also acknowledge that the absence of a comparative condition limits the internal validity of our findings, and that the homogeneity of the sample may constrain the external validity and generalizability of the results to more diverse educational contexts. Future studies could benefit from implementing quasi-experimental designs with matched control groups and including participants from a broader range of school settings, in order to better account for contextual variation and enhance the robustness of the conclusions
Reviewer: Finally, the bibliography is largely adequate and recent, though somewhat insular. Many of the cited works are authored by the research team or collaborators. While this is understandable in niche fields, including a broader set of international references could strengthen the manuscript’s positioning within the global literature.
Authors answer: Additional references have been incorporated into the study, among them Mancini et al. (2024).
Reviewer: In conclusion, this study offers a valuable contribution to the understanding of how fundamental and sport-specific motor skills develop over time. The intervention is well-conceived and the longitudinal approach is particularly commendable. However, revisions in the theoretical framework, statistical interpretation, and stylistic clarity—as well as the integration of new literature such as Mancini et al. (2024)—are recommended to enhance the manuscript's impact and rigor. With these improvements, the article will better serve the field of developmental motor behavior and physical education pedagogy.
Authors answer: Thank you for your highly generous comments and suggestions that enhanced the new draft, where many changes have been made. We truly hope that it now meets the quality standards for publication in Sports.

Reviewer 2 Report
Comments and Suggestions for Authors
I found this manuscript to be both relevant and methodologically thoughtful. The authors address an important question in developmental motor science: the temporal relationship between fundamental movement skills (FMS) and sport-specific skills in children. Their longitudinal design, spanning four data collection waves over a two-year period, demonstrates a commendable commitment to capturing developmental change. I particularly appreciated the dual-path modeling strategy, which allowed the authors to investigate both concurrent and lagged effects—an analytical choice that reflects a nuanced understanding of motor learning as a process that unfolds over time.
That said, while I was impressed by the rigor of the data collection and the intervention’s ecological validity, several aspects of the manuscript could be further strengthened. At times, the argument felt constrained by a descriptive tone, particularly in the Results and Discussion sections, where opportunities to theorize more boldly were missed. Furthermore, although the models were well-conceived, I was concerned by the marginal fit indices and felt that this issue deserved deeper reflection in the interpretation.
The manuscript would also benefit from a more transparent editorial structure and better articulation of the theoretical stakes of the study. Overall, this is a valuable contribution, but I recommend a series of targeted revisions before publication.
Specific Comments
Lines 14–26
Upon reading the abstract, I found the language to be unnecessarily dense. Phrases like “time-lagged influence” appeared without sufficient framing, and the emphasis on statistical significance overwhelmed the reader before the conceptual contribution had been established. I recommend simplifying and sharpening the language to make the study’s main finding—the delayed influence of FMS on sport skills—more digestible at a glance.
Lines 42–44:
The example of combining running and dribbling into more complex tasks, such as a basketball layup, was helpful. Still, I would have liked to see more reflection on the developmental mechanisms involved. The term “elaborate structures” remains abstract. I encourage the authors to clarify what kinds of cognitive or biomechanical integration are implied by this statement.
Lines 87–91
I appreciated the articulation of the three research questions, but I found that question (ii) overlapped conceptually with (i) and (iii). As a reader, I had difficulty discerning how it was analytically distinct. Rephrasing (ii) to focus more explicitly on the role of intervention design (e.g., timing, content specificity) might improve the clarity of this section.
Lines 107–125
I value the authors’ commitment to ecological validity—the decision to embed the intervention in school PE lessons is well justified. However, I noticed the manuscript is silent on fidelity monitoring. Was there any qualitative or observational data on how consistently the PE teacher implemented the intervention? This omission left me questioning the internal consistency of the pedagogical delivery.
Lines 140–151
Acknowledging a potential ceiling effect in running is essential, especially when interpreting longitudinal change. However, this issue is mentioned only in passing. Given that the modeling depends on the variability of change, the ceiling effect deserves more systematic attention. I would like to know whether this led to the underestimation of specific effects.
Lines 221–226
Table 1 initially confused me, as it listed “Running and Stationary Dribbling 1” four times. I assume this is a formatting error and that these rows were meant to represent time points 1 through 4. Cleaning up this presentation would greatly enhance readability.
Lines 238–240
The authors are commendably transparent about their model fit statistics. Yet, I was surprised by how little interpretive weight they gave to the fact that the RMSEA exceeds acceptable thresholds (RMSEA = 0.124). I would have appreciated a more critical discussion of what this means for the validity of the concurrent-effect hypothesis.
Lines 311–316
The emergence of significant associations only between RD3 and Db4 struck me as the most interesting finding in the manuscript. However, the authors stop short of speculating more deeply on why this lag occurs. Could it be related to consolidation processes or variability-induced adaptability? Referencing dynamical systems or Newell’s constraints theory here could have enriched this interpretation.
Lines 333–356
The authors acknowledge the lack of a control group and the limited scope of FMS skills assessed, but their discussion feels defensive rather than reflective. I would have preferred a more open discussion of how these limitations constrain the generalizability of findings. At the same time, I recognize the value of ecological validity in real-world physical education (PE) settings and appreciate the authors’ sensitivity to school demands.
Lines 380–460
The references are current and appropriate, and I was pleased to see work by Stodden and Seefeldt cited. Still, I found the theoretical framework too reliant on linear stage models. Including perspectives from ecological dynamics or skill acquisition literature could help bridge the gap between theory and the developmental delays observed.
Author Response
Reviewer 2
Comments and Suggestions for Authors
I found this manuscript to be both relevant and methodologically thoughtful. The authors address an important question in developmental motor science: the temporal relationship between fundamental movement skills (FMS) and sport-specific skills in children. Their longitudinal design, spanning four data collection waves over a two-year period, demonstrates a commendable commitment to capturing developmental change. I particularly appreciated the dual-path modeling strategy, which allowed the authors to investigate both concurrent and lagged effects—an analytical choice that reflects a nuanced understanding of motor learning as a process that unfolds over time.
That said, while I was impressed by the rigor of the data collection and the intervention’s ecological validity, several aspects of the manuscript could be further strengthened. At times, the argument felt constrained by a descriptive tone, particularly in the Results and Discussion sections, where opportunities to theorize more boldly were missed. Furthermore, although the models were well-conceived, I was concerned by the marginal fit indices and felt that this issue deserved deeper reflection in the interpretation.
The manuscript would also benefit from a more transparent editorial structure and better articulation of the theoretical stakes of the study. Overall, this is a valuable contribution, but I recommend a series of targeted revisions before publication.
Authors answer: We want to thank the reviewer for their very generous, interesting, and challenging comments regarding our paper. We also thank her/him for the very positive impression of our study.
Specific Comments
Reviewer: Lines 14–26
Upon reading the abstract, I found the language to be unnecessarily dense. Phrases like “time-lagged influence” appeared without sufficient framing, and the emphasis on statistical significance overwhelmed the reader before the conceptual contribution had been established. I recommend simplifying and sharpening the language to make the study’s main finding—the delayed influence of FMS on sport skills—more digestible at a glance.
Authors answer: Thank you for the suggestion. We simplified the abstract to improve readability and clarified the emphasis on the main finding—the delayed influence of FMS on sport skills (now abbreviated as SSS).
Reviewer: Lines 42–44:
The example of combining running and dribbling into more complex tasks, such as a basketball layup, was helpful. Still, I would have liked to see more reflection on the developmental mechanisms involved. The term “elaborate structures” remains abstract. I encourage the authors to clarify what kinds of cognitive or biomechanical integration are implied by this statement.
Authors answer: Thank you for the recommendation. We incorporated additional theoretical background in the Introduction to clarify the developmental mechanisms and provide a stronger foundation for our research question.
The following was added:
Despite the relevance of descriptive models in the field of motor development, most are grounded in sequential perspectives and lack explicit reference to the mechanisms underlying change. One proposal that helps conceptualize these mechanisms is offered by Tani (2005), who describes motor development as a hierarchically organized process aimed at forming increasingly complex motor patterns. Within this framework, the acquisition of fundamental movement skills (FMS) functions as a basic motor patterns that, over time, become components for the acquisition of more complex motor actions, such as SSS.
For this progression to occur, the child must be able to adjust key motor parameters of the motor patterns—such as force, speed, and direction—without compromising their efficiency. This flexibility enables the functional dismantling of the original FMS, allowing them to be combined into more sophisticated motor patterns.
For example, once a child masters the FMS of running and bouncing a ball, these motor patterns can be used as components of more complex motor patterns, such as speed dribbling. Importantly, this hierarchically organized process is not a simple additive process of components. It involves a reorganization process in which the flexibility of the components plays a crucial role. As the motor development unfolds, what was once a whole (e.g., speed dribbling) may later become a component of an even more complex motor pattern, such as executing a basketball layup. In this hierarchical process, the whole becomes a part, and parts are reorganized to form new wholes (Koestler, 1967).
Crucially, this hierarchical process is far from trivial. Using FMS as building blocks for more elaborate skills requires flexibility. The learner must be able to modify performance parameters (e.g., force, speed, and direction) and adapt each component of the FMS to accommodate new task demands. In other words, the motor pattern of the original FMS must be partially "deconstructed" or altered to allow for functional reorganization into new motor patterns. Thus, the progression from FMS to SSS is better understood as a process of increasing behavioral diversity (expanding the range of motor elements by parameters modification) and complexity (integrating these elements into cohesive new patterns) (Tani, 2012).
In this context, both the passage of time and the level of proficiency in FMS appear to be critical factors when examining their influence on the development of SSS. However, as highlighted in a recent narrative review by Garbeloto and Pereira (2024), the relationship between FMS and SSS remains largely underexplored. The review identified only five studies that have directly investigated the connection between FMS and SSS, indicating a significant gap in the literature, particularly regarding how this relationship unfolds in time.
Reviewer: Lines 87–91
I appreciated the articulation of the three research questions, but I found that question (ii) overlapped conceptually with (i) and (iii). As a reader, I had difficulty discerning how it was analytically distinct. Rephrasing (ii) to focus more explicitly on the role of intervention design (e.g., timing, content specificity) might improve the clarity of this section.
Authors answer: Again, thank you for the suggestion. We revised research question (ii) to emphasize the role of intervention design, which helped clarify the distinction between the three questions.
The following was added:
(ii) How do specific features of the intervention program—such as its duration, content focus, and integration into regular PE lessons—influence the dynamics between FMS and SSS development?
Reviewer: Lines 107–125
I value the authors’ commitment to ecological validity—the decision to embed the intervention in school PE lessons is well justified. However, I noticed the manuscript is silent on fidelity monitoring. Was there any qualitative or observational data on how consistently the PE teacher implemented the intervention? This omission left me questioning the internal consistency of the pedagogical delivery.
Authors answer: We acknowledge the importance of intervention fidelity and appreciate the reviewer’s generous remark to this point. To monitor pedagogical consistency, the lead researcher attended three of the ten sessions—specifically, the first, fourth, seventh, and tenth classes—to verify adherence to the planned instructional procedures. Additionally, the PE teacher was instructed to contact the lead researcher in case any session deviated from the original plan or if any uncertainties arose during implementation. This process ensured a basic level of fidelity monitoring, despite the naturalistic setting of the intervention. This information was added to the main text.
The following was added:
To monitor pedagogical consistency, the lead researcher attended four of the ten sessions—specifically, the first, fourth, seventh, and tenth classes—to verify adherence to the planned instructional procedures. Additionally, the PE teacher was instructed to contact the lead researcher in case any session deviated from the original plan or if any uncertainties arose during implementation.
Reviewer: Lines 140–151
Acknowledging a potential ceiling effect in running is essential, especially when interpreting longitudinal change. However, this issue is mentioned only in passing. Given that the modeling depends on the variability of change, the ceiling effect deserves more systematic attention. I would like to know whether this led to the underestimation of specific effects.
Authors answer: We appreciate the reviewer’s observation regarding the potential ceiling effect in the running skill, and we agree that this issue merits more systematic consideration. In our dataset, several children scored at or near the maximum performance level for running immediately after the intervention, which likely reduced the variability needed for detecting further change in subsequent assessments. This restriction in range may have contributed to the attenuation of certain effects within the path models, particularly those involving longitudinal associations between FMS and sport-specific skills.
The following was added:
Another limitation concerns the focus on only two FMS and one SSS, which may have resulted in some children reaching a performance ceiling, particularly in running, shortly after the intervention. Given that our model depends on individual differences in change over time, a ceiling effect could indeed lead to an underestimation of the influence of running proficiency, both in isolation and in combination with other FMS (e.g., stationary dribbling) when predicting SSS outcomes. Although the combined FMS score (running + stationary dribbling) helped mitigate this issue to some extent, we recognize that running may have disproportionately contributed to the reduced variance in later waves.
Please note also that we explicitly address this limitation in the discussion section and encourage future studies to consider alternative scoring systems or more challenging assessment protocols to better capture variability in highly proficient individuals.
Reviewer: Lines 221–226
Table 1 initially confused me, as it listed “Running and Stationary Dribbling 1” four times. I assume this is a formatting error and that these rows were meant to represent time points 1 through 4. Cleaning up this presentation would greatly enhance readability.
Author answer: Many thanks for this remark. There was indeed a formatting error in Table 1 where “Running and Stationary Dribbling 1” was listed four times. We have corrected this issue by properly labeling the rows to represent time points 1 through 4.
Reviewer: Lines 238–240
The authors are commendably transparent about their model fit statistics. Yet, I was surprised by how little interpretive weight they gave to the fact that the RMSEA exceeds acceptable thresholds (RMSEA = 0.124). I would have appreciated a more critical discussion of what this means for the validity of the concurrent-effect hypothesis.
Authors answer: We thank the reviewer for issue which was also raised by Reviewer 1. In fact, given the small sample size and the time lag between observations, the RMSEA is within the boundary of a reasonable fit, even using a robust estimation procedure. Please note, however, that we followed the general “rules” regarding these types of models and tried to maintain a perspective of model testing, and not follow the ideas of model re-specification that sometimes tend to overparameterize models without a substantive reason. EQS has a series of capabilities regarding model changes. We used them and found that many suggestions were impractical given what the software names “condition code”, i.e., negative variances and/or correlations greater than 1, and others would change the main aim of both models. These are the reasons why we “stick” to our model testing. Please note also that both models had the same degrees of freedom and the same value of Satorra-Bentler Chi-square. This way AIC or BIC would be the same, and so we did not use it.
Please note, however, that path models deal with variances and covariances, not means. To test for gains, linear or curvilinear, across the four data waves would require a different statistical approach, and to test for the presence of environmental factors as well as putative mediating variables would be something beyond the reach of this paper, and we wonder if there is a published paper that ever tackled such a complex issue. In any case, we add a “word” about this.
We add a comment on these issues within the lines of our comments.
Our overall model fit shows values within the boundary of acceptable fit in these types of models with direct observed variables. This may be linked to a small sample size, a different time lag between data waves, and/or other environmental issues that were not considered in the study. It is also important to note that in these types of models only variances and covariances are used (stability and coincident/cross-lagged effects), and do not address changes in means across time. We were very careful in not relying on model re-specification freeing parameters that were not substantively interpretable even at the cost of increasing model fit statistics. Hence, and within the limits of these considerations, we used a robust estimation procedure to tackle the issue of the sample size, and have precise standard-errors which are important for significant testing of all regression effects, correlations, and error covariances. Further, and since both models have the same number of degrees of freedom and the same chi-square statistic and other fit indices (CFI and RMSEA), we believe that they can be suitable interpretable within the study framework, and the choice depends on how much different they are in substantive terms, and/or what in what they suggest in terms of transitional influence of FMS on SS
Reviewer: Lines 311–316
The emergence of significant associations only between RD3 and Db4 struck me as the most interesting finding in the manuscript. However, the authors stop short of speculating more deeply on why this lag occurs. Could it be related to consolidation processes or variability-induced adaptability? Referencing dynamical systems or Newell’s constraints theory here could have enriched this interpretation.
Authors answer: We thank the reviewer for the insightful comment and the suggestion to explore possible explanations for the observed lag between RD3 and Db4.
The following was added:
This process aligns with the theoretical perspective of Tani et al. (1995), who proposed that, from a hierarchical view of motor development, FMS function as motor patterns that, over time, become basic components for the acquisition of more complex skills, such as SSS. For this progression to occur, the child must be able to adjust key motor parameters—such as force, speed, and direction—without compromising the efficiency of motor patterns. This flexibility enables a functional dismantling of the original FMS, allowing their components to be reorganized into more sophisticated motor patterns.
Based on the hierarchical approach, this process illustrates how elements from running and stationary movement patterns were first consolidated through practice and subsequently reorganized to form a more complex skill—speed dribbling in basketball. This dynamic reorganization, in which motor patterns alternate between functioning as autonomous skills and as building blocks of more advanced motor actions, depends on sufficient time and practice conditions. In our intervention program, these conditions—specifically, the promotion of movement variability and the progressive increase in task complexity—were deliberately integrated to support this developmental progression.
We would also like to acknowledge the importance of Newell’s constraints model in explaining individual variability in motor development. In fact, we have applied this model in the context of a separate study (please see the reference below), where it served as the foundation for a new pedagogical approach that highlights how children may progress at different rates depending on the interaction of individual, task, and environmental constraints.
However, while we greatly value Newell’s contribution to the field and recognize the potential of the constraints model to enrich our understanding of individual variability in motor development, our theoretical approach in the present manuscript is based on a distinct framework—namely, a hierarchical perspective of motor development. This choice was guided by the structure of our intervention, which emphasized sequential skill progression and reorganization, rather than adaptive variability per se. For this reason, we chose not to integrate the constraints model directly into this article, as it follows a different conceptual trajectory.
Reference
Pereira, S.; Santos, C.;Maia, J.; Vasconcelos, O.;Guimarães, E.; Garganta, R.;Farias, C.; Barreira, T.V.; Tani, G.;Katzmarzyk, P.T.; et al. Children’s Individual Differences in the Responses to a New Method for Physical Education. Sports 2024, 12,328. https://doi.org/10.3390/sports12120328
Reviewer: Lines 333–356
The authors acknowledge the lack of a control group and the limited scope of FMS skills assessed, but their discussion feels defensive rather than reflective. I would have preferred a more open discussion of how these limitations constrain the generalizability of findings. At the same time, I recognize the value of ecological validity in real-world physical education (PE) settings and appreciate the authors’ sensitivity to school demands.
Authors answer. We thank both reviewers for their constructive feedback regarding the limitations section. In response, we have revised this part of the manuscript to adopt a more reflective tone and to openly discuss how the absence of a control group, the limited scope of motor skills assessed, and the sample's homogeneity may constrain the generalizability of our findings. We have also acknowledged how these factors impact both internal and external validity. At the same time, we continue to emphasize the ecological validity of conducting research in authentic school settings. We hope this revised discussion more clearly balances the practical constraints of real-world implementation with a transparent account of methodological limitations.
Reviewer: Lines 380–460
The references are current and appropriate, and I was pleased to see work by Stodden and Seefeldt cited. Still, I found the theoretical framework too reliant on linear stage models. Including perspectives from ecological dynamics or skill acquisition literature could help bridge the gap between theory and the developmental delays observed.
Authors answer: We thank the reviewer for the thoughtful comment and for recognizing the relevance of the references cited. In response to your suggestion, we have expanded the theoretical framework to include the perspective of hierarchical motor development, as proposed by Tani et al. (1995), to offer a deeper explanation for the delayed effects observed in our findings. This approach allows us to conceptualize how fundamental movement skills (FMS) function as action programs that become subroutines for more complex skills over time, providing a theoretical basis for understanding why certain associations emerge only at later stages.
While we appreciate the value of ecological dynamics and the skill acquisition literature—particularly regarding variability and adaptability—our study is grounded in a different conceptual framework, which emphasizes hierarchical organization rather than dynamic systems. We believe that including this additional theoretical layer strengthens the manuscript by bridging the gap between developmental theory and the observed empirical patterns.
